# OpenReview forum: "Biological Sequence Editing with Generative Flow Networks"
_ICLR.cc/2024/Conference — Submitted to ICLR 2024_

### Official Review · Reviewer_Arqn · 2023-10-28

**Soundness:** 3 good
**Presentation:** 3 good
**Contribution:** 3 good
**Rating:** 6
**Confidence:** 4

**Summary:**

This submission isn concerned with the application of GFNs to sequence modifications. I’m not up to date with the relatively large number of works on similar approaches. The related work section appears in the supplementary material and does not attempt to discuss the differences between this paper and the papers listed there. This is a key issue that most be resolved. Also there is no comparison with earlier GFN methods: is there any for these problems? These concerns have made me give a lower score than what I otherwise would have given. I have minor comments and questions below, but I really enjoined reading this submission. It provides a description of the method at hand as well as the general GFN approach on very well chosen level. It is mostly well written. Moreover, the approach make sense and the results are good. I would like to accept it, but the above issue should be resolved.

**Strengths:**

Well written, technically strong, good resuls.

**Weaknesses:**

Poorly described relation to previous research and unclear if compared with the right methods.

**Questions:**

* that can provide consistency condition, Bengio et al. (2021) formulates ow- matching loss function as follows: LFM(s; θ) = log P∀s′:s′→s Fθ(s′ → s) P′′′′ Fθ(s → s′′) . (4) Moreover, as an alternative objective function  Page 3  Sure but which do you use?
* 2  R(x) Page 3  Not defined here.
* GFlowNet’s ow function Fθ(·) to identify sub-optimal positions of x, and subsequently replace the sub-optimal parts with newly sampled edits based on the stochastic policy π(·). pretrained ow function Fθ(·) Page 4  At this point its implementation is not clear or why it can be pre trained
* For instance for the DNA sequence x = ‘ATGTCCGC’, appending token a = ‘C’ to x:2, we get x:2 + a = ‘ATC’.  Page 4  I would guess that this is an insert operation on x, but it is not clear from the description, which actually suggests that the suffix from position t+1 is removed from x. New guess: you are stepwise building a sequence and you either use the character from the given sequence or another. You should make this more clear.
* 5  can Page 4  Would
* 6  Pa′∈A Page 4  It should be made clear that the given character x_t always belongs to the available actions.
* 7  chosen by the algorithm Page 4  Point out how or, alternatively, where you describe it.
* 8  regularization parameter λ allows tuning Page 5  Point out how it is set or where you describe it.
* 9  RF,T represents the reward of a sequence with length T generated using the ow function Fθ(·)  Page 5  It IS the reward. Formulate it as a rv with the distribution induced by the flow function, with reference to the correct equation.
* 10  Levenshtein  Page 6  Spelling.

* Higher binding activity is preferable Page 6  In what sense? It may not be so in a biological system.
* 12  diversity Page 7  You need a better measure that also takes the improvement into account.
* 13  14.34 Page 7  This should be in bold, right?

---

> ### Author Response · Authors · 2023-11-17
> **Responses to comments**
>
> We sincerely appreciate your thoughtful review of our manuscript. In response to your feedback, we meticulously revised the introduction, conducted additional experiments, and added new experimental results to strengthen our paper. Please find below a detailed point-by-point response addressing your comments.
>
> ## Related Works
> Existing machine learning methods within the domain of biological sequences including the ones discussed in the related works have predominantly concentrated on generating novel *de novo* sequences with desired properties from scratch. Therefore, there is an inherent risk of deviating significantly from naturally occurring sequences compromising safety and predictability. In contrast, this paper explores the generation of edited sequences that enhance the properties of existing sequences, ensuring the edited counterparts maintain similarity to their original counterparts. Here, we take an input seed sequence and modify a few elements to improve its properties.
>
> To underscore the limitations of existing biological sequence design methods, we conducted experiments on AMP and CRE datasets using GFlowNet to generate sequences from scratch. Our observations reveal that GFlowNet-generated sequences for the CRE dataset differ from existing sequences at 76.29% of locations, and for the AMP dataset, the difference is found in 95.24% of locations. In contrast, the proposed GFNSeqEditor demonstrates the ability to generate new edited sequences differing from existing ones in less than 35% of locations across all datasets. Moreover, employing existing biological sequence methods the difference between generated sequences and existing ones cannot be controlled while using the proposed GFNSeqEditor the amount of edits can be controlled by hyperparameters.
>
> In order to highlight the differences between biological sequence editing and biological sequence design problems we have added a discussion to the introduction of the revised manuscript. This revision can be found on pages 1 and 2 highlighted in red. Also, we extend the related works discussion in Appendix D and the revisions are highlighted in red.
>
> Furthermore, another aspect that distinguishes the proposed GFNSeqEditor from prior works is that existing methods are not able to combine multiple sequences to create a new sequence. In section 4.3, the proposed algorithm combines sequences to generate a new sequence similar to its parent sequences. In section 4.3, we discuss that this can also be used for sequence length reduction which has important applications in vaccine therapies.
>
> ## Comparison with GFN Methods
> Existing GFlowNet methods can be used for biological sequence design from scratch and as a result, they suffer from the same problems as those of other biological sequence design methods discussed above when it comes to performing sequence editing. However, since GFlowNet can generate sequences token by token sequentially, there is a naive approach to employ GFlowNet for sequence editing. In this approach, the first tokens of a sequence $x$ can be given to GFlowNet and then the GFlowNet generates the rest of the tokens. We add this as a baseline to Table 1 in the revised version and the baseline is called GFlowNet-E. For example, for the CRE dataset, the GFlowNet-E gets the first 130 tokens and generates the rest of 70 tokens. Compared to GFNSeqEditor, this approach lacks the flexibility to choose edit locations and GFNSeqEditor outperformed GFlowNet-E. This indicates the effectiveness of sub-optimal position identification of GFNSeqEditor.

---

> > ### Comment · Reviewer_Arqn · 2023-11-21
> > **Related works and comparison**
> >
> > I find these new texts well-written, clarifying, and clearly adding to the value of the submission.

---

> > > ### Author Response · Authors · 2023-11-22
> > >
> > > We appreciate the positive feedback regarding the clarity and value of our response. As we approach the end of the discussion period, we are taking this opportunity to inquire if you have any specific concerns or queries regarding our paper. If our revised paper and responses have addressed your concerns, we ask you to kindly consider increasing your score. Thank you again for helping us to improve our paper.

---

> ### Author Response · Authors · 2023-11-17
> **Responses to Questions**
>
> Please find below our responses to your questions.
>
> 1. We used trajectory balance loss in equation (5) to train the flow function. On page 7 before section 4.1 of the revised version, we made clear that we used trajectory balance loss. The revision is highlighted in red.
> 2. In general the reward function $R(x)$ can be any function and it is determined by the environment. In experiments we consider the sequence property as the reward since in experiments our goal was to maximize those properties.
> 3. In the first paragraph of section 3 we add that “Assume that the flow function $F_\theta(\cdot)$ is trained on an available offline training data.”
> 4. In order to make the sequence editing procedure of GFNSeqEditor more clear, in section 3.1 we add that “GFNSeqEditor constructs edited sequences token by token and for each position $t+1$ it examines if $x_{t+1}$ should be used or not.”
> 5. We revised can to would.
> 6. We add a comment on page 5 that “Assume that $x_t \in \mathbb A$, $\forall t$ meaning that $x_t$ is always in the available actions.”
> 7. To clarify how to choose hyperparameters $\delta$, in this revised version after equation (6) we add that “Choosing larger $\delta$, it is more probable that the algorithm identifies $x_{t+1}$ as sub-optimal”. Furthermore, to clarify how to choose hyperparameter $\sigma$ after equation (7) we add that “The relation between $\sigma$ and the algorithm performance will be analyzed in section 3.3 and Appendix E.”
> 8. In order to clarify the effect of $\lambda$ in this revised version after equation (9), we add “Choosing larger $\lambda$ leads to obtaining smaller number of edits”.
> 9. Based on this comment, we revise the notation and in the revised version $R_{F,T}$ denotes the expected reward of a sequence with length $T$ generated using the flow function $F_\theta(\cdot)$.
> 10. We checked the spelling and we believe it is correct.
> 11. We followed the prior studies such as Trabucco et al., (2021) and Jain et al., (2022) in which the goal in the TFbinding dataset is to increase the binding activity.
> 12. We would like to clarify that diversity itself is an important measure in generating biological sequences. Higher diversity as it is defined in section 4 among edited sequences increases the chance that at least one of the edited sequences successfully works in real experiments since it is expected that similar sequences show somewhat similar behavior in real experiments. Furthermore, diversity defined on page 7 is used before to evaluate the performance of biological sequence design methods (see e.g. Jain et al., (2022)).
> 13. We made14.34 bold in Table 1.

---

### Official Review · Reviewer_aGg2 · 2023-10-30

**Soundness:** 3 good
**Presentation:** 2 fair
**Contribution:** 3 good
**Rating:** 6
**Confidence:** 3

**Summary:**

This paper introduces an innovative algorithm called GFNSeqEditor, which is specifically crafted to enhance sequences by optimizing their desired properties. GFNSeqEditor harnesses the power of pretrained flow functions and devises a set of destructive operations, whether tokens are modified or not. It then reconstructs the altered tokens using these pretrained flow functions. The process of destruction and subsequent reconstruction is governed by three crucial hyperparameters: $\lambda$, $\alpha$, and $\delta$. These hyperparameters play a pivotal role in achieving a balance between exploration and exploitation, while also managing the trade-offs and risks associated with expected improvements.

The authors provide a comprehensive analysis of these proposed hyperparameters, which intuitively guide the algorithm's behavior. To evaluate its effectiveness, GFNSeqEditor is benchmarked against classical editing methods across three distinct sequential generation tasks.

**Strengths:**

This paper excels in storytelling, skillfully introducing a promising generative model for sequence editing. The approach itself is novel and the underlying concept is commendable. The subsequent algorithm, while simple, remains straightforward, and the mathematical analysis of the newly introduced hyperparameters is intuitively presented. Overall, this paper is highly accessible and a pleasure to read.

**Weaknesses:**

The primary weakness of this paper lies in its experimental validation. In my opinion, the experiments conducted here fall short of adequately substantiating the proposed idea. There are several issues that need addressing:

**Pretraining Discrepancy**: One notable concern is the difference in the starting points for experimentation. While this work leverages pretrained GFN models, other baselines begin from scratch. This discrepancy could potentially lead to an unfair comparison.

**Baseline Variety**: The baseline comparisons should extend beyond the scope of other generative models and optimization techniques in biological sequence design. It would be beneficial to incorporate baseline methods such as offline model-based optimization [1], which are tailored to extrapolate sequences from offline datasets, thereby yielding "improved sequences."

**Evolutionary Algorithms**: To provide a more comprehensive perspective on the proposed approach, the paper could benefit from the inclusion of promising evolutionary algorithms specifically designed for biological sequence design [2].

**Comparison with GFN Baselines**: Additionally, conducting a thorough comparison with GFN baselines would be valuable in demonstrating the relative strengths and weaknesses of the proposed method when contrasted with models of similar architecture.

Addressing these concerns would significantly enhance the rigor and completeness of the experimental evaluation in the paper.

[1] Trabucco, Brandon, et al. "Design-bench: Benchmarks for data-driven offline model-based optimization." International Conference on Machine Learning. PMLR, 2022.

[2] Sinai, Sam, et al. "AdaLead: A simple and robust adaptive greedy search algorithm for sequence design." arXiv preprint arXiv:2010.02141 (2020).

**Questions:**

1. Could you please elaborate on the process you used for pre-training GFN?

2. Have you conducted a comparison with baseline models (e.g., Seq-to-Seq) using the pretrained GFN as a component?

3. Does this algorithm demonstrate improvements in scalability?

4. Is this algorithm more beneficial than naive search algorithms based on pretrained policies, such as beam-search or MCTS?

5. Have you performed experiments related to Theorem 1 and 2? Inclusion of such experiments would likely enhance the overall quality of the paper.

---

> ### Author Response · Authors · 2023-11-17
> **Response to Comments**
>
> We would like to thank you for taking time to review our paper and let us know your valuable comments. Based on your comments, we have added more discussions on related works to the introduction and enriched the experiments section with additional experimental results. Kindly find below a detailed point-by-point response addressing each of your comments.
>
> ## Pretraining
> We would like to clarify that other baselines also use a pretrained model to perform editing. While we trained GFlowNet for GFNSeqEditor, we trained an MLP as the proxy model for Ledidi and Directed Evolution (DE) baselines. Ledidi and DE are highly dependent on the proxy model to evaluate their candidate edits and use these evaluations to proceed to the next iteration for editing. Also, we trained a Transformer for the Seq2Seq baseline and Seq2Seq employs the trained transformer model to perform sequence editing. Please note that:
> 1. We used exactly the same training data to pretrain GFlowNet and models used by the other baselines.
> 2. Due to the space limit, we put the experimental details in Appendix C.1.
>
> Therefore, we believe that comparisons in the paper are fair since all baselines use a trained model and all models are trained on the same training data. Thanks to this comment to improve the clarity of the experiment section, we expand implementation details explanations in Appendix C.1 and also in section 4 we refer the readers to Appendix C.1.
>
> ## Baselines
> We would like to clarify that biological sequence design methods such as offline model-based optimization models including the ones discussed in [1] do not tackle the problem of sequence **editing** and as a result, they are not appropriate to be directly used for biological sequence editing for two main reasons:
> 1. Biological sequence design models usually generate entirely new sequences from scratch. Therefore, sequences generated by these methods are expected to be greatly different from existing sequences that need editing. Therefore, it might be infeasible to generate diverse sequences using biological sequence design models that are similar to the existing sequences.
> 2. The difference between generated sequences and existing ones cannot be controlled by employing biological sequence design methods while using the proposed algorithm by setting hyperparameters one can control the number of edits.
>
> To underscore the limitations of biological sequence design methods, we generate new sequences from scratch using GFlowNet and we observe that on average for CRE dataset the generated sequences are different from existing sequences (i.e. sequences in the datset) in 76.29% of locations and it is not possible to control the amount of differences between generated sequences and existing ones. Also, for AMP dataset the generated sequences are different from existing ones in 95.24% of locations. In contrast, the proposed GFNSeqEditor demonstrates the ability to generate new edited sequences differing from existing ones in less than 35% of locations across all datasets. To highlight the differences between biological sequence editing and biological sequence design problems we have added a discussion to the introduction of the revised manuscript. The revision can be found on pages 1 and 2 highlighted in red.
>
> ## Evolutionary Algorithms
> Thanks for drawing our attention to the evolutionary algorithm in [2]. Note that [2] implements a standard directed evolutionary algorithm for sequence design. Our directed evolutionary baseline is the same as the evolutionary algorithm presented in Algorithm 1 of [2]. We cite the reference [2] in the revised version.
>
> ## Comparison with GFN Baselines
> Existing GFN baselines can be used for biological sequence design from scratch and as a result, they suffer from the same problems as those of other biological sequence design methods discussed above when it comes to performing sequence editing. However, since GFlowNet can generate sequences token by token sequentially, there is a naive approach to employ GFlowNet for sequence editing. In this approach, the first tokens of a sequence $x$ can be given to GFlowNet and then the GFlowNet generates the rest of the tokens. We add this as a baseline to Table 1 in the revised version and the baseline is called GFlowNet-E. For example, for the CRE dataset, the GFlowNet-E gets the first 130 tokens and generates the rest of 70 tokens. Compared to GFNSeqEditor, this approach lacks the flexibility to choose edit locations. GFNSeqEditor outperforms GFlowNet-E and this indicates the effectiveness of sub-optimal position identification of GFNSeqEditor in equation (7).

---

> ### Author Response · Authors · 2023-11-17
> **Responses to Questions**
>
> Please find below our responses to you questions.
>
> **Response to Q1.** We trained an active learning based GFlowNet model following the setting in Jain et al. (2022). In the active learning setting, at each round of active learning $t \times K$ candidates generated by GFlowNet are sampled and then top $K$ samples based on scores given by a proxy are chosen to be added to the offline dataset. Here offline dataset refers to an initial labeled dataset. To train the GFlowNet, we employed the same proxy models as those used by other baseline methods. For all datasets, we set the number of active learning rounds to $1$, with $t$ equal to $5$ and $K$ equal to $100$. We parameterize the flow using a MLP comprising two hidden layers, each with a dimension of $2048$, and $|\mathbb A|$ outputs corresponding to individual actions. Throughout our experiments, we employ the trajectory balance objective for training. Adam optimizer with $(\beta_0, \beta_1) = (0.9, 0.999)$ is utilized during the training process. The learning rate for $\log Z$ in trajectory balance loss is set to $10^{-3}$ for all the experiments. The number of training steps for TFbinding, AMP and CRE are $5000$, $10^6$ and $10^4$, respectively. The remaining hyperparameters were configured in accordance with the settings established in Jain et al. (2022).
>
> Due to space limitations, we provided the training details of employed GFlowNet models in Appendix C.1. In this revised version, we provided more training details in the main text of the paper. This revision can be found on page 7 before section 4.1 highlighted in red.
>
> **Response to Q2.** Our baselines in the initial submission do not use the trained GFN since they do not construct sequences token by token sequentially. However, all baselines perform editing using a trained model. All models including GFN and the transformer of Seq2Seq are trained on the same training data.
>
> **Response to Q3.** The proposed algorithm needs to query for the flow of partially constructed sequences $|\mathbb A|T$ times where $|\mathbb A|$ is the size of the action set (which is $4$ for TFbinding and CRE datasets and is $20$ for AMP dataset) and $T$ is the length of the sequence. Querying the evaluation of trained flow function models only needs inference and does not involve training. As a result, querying the evaluation of trained flow function is not computationally complex. Therefore, the complexity of the algorithm linearly scales with the size of the action set and the length of the input sequence. Therefore, the algorithm is scalable.
>
> **Response to Q4.** Generating a new sequence from scratch by GFlowNet can be viewed as a search algorithm based on a pretrained policy. The benefit of using the learned policy by GFlowNet compared to other policies is that the probability of constructing a sequence by GFlowNet is proportional to the sequence reward. Moreover, naive search algorithms such as beam-search and MCTS usually need to observe the intermediate reward. However, when it comes to constructing a new object sequentially, the intermediate reward is not available and the reward is given when the object is fully constructed. Therefore, naive search algorithms may not work well in generating new objects. GFlowNet is able to learn to generate a new object with probability proportional to its final reward when there is not any intermediate reward. Leveraging GFlowNet by the proposed algorithm is its benefit compared to naive search algorithms.
>
> **Response to Q5.** We performed experiments related to Theorems 1 and 2. Figure 3 studies the effect of hyperparameters $\delta$ and $\lambda$ on the property  improvement and edit percentage. Furthermore, in Figure 4, we illustrate the impact of changing $\sigma$ on property improvement and edit diversity for GFNSeqEditor. These results corroborate the theoretical analyses outlined in Theorems 1 and 2 in section 3.3 as well as Theorem 3 in Appendix E.

---

> ### Comment · Reviewer_aGg2 · 2023-11-20
> **Potentially nice work, but refine the paper further in next time**
>
> The rebuttal and the improved manuscript have indeed addressed some of my concerns.
>
> However, I hold a different perspective on the statement, "Moreover, naive search algorithms such as beam-search and MCTS usually need to observe the intermediate reward." I believe that we can leverage a Value estimator to facilitate beam-search or MCTS without requiring access to the true reward, similar to the approach used in AlphaGo.
>
> It's important to note that there are numerous ways to generate edit sequences. To bolster your method's credibility, I recommend exploring a wide array of scenarios to showcase its superiority over various alternatives.
>
> Furthermore, while I do partially agree with your assertion that editing methods can outperform de novo methods in terms of novelty (deviation from current sequences), I believe that this comparison should be made more explicit in the paper. Specifically, it would be beneficial to include (1) the actual metric used for comparison and (2) a comparison with baseline de novo methods (I appreciate you providing some experiments on this rebuttal). Doing so will enhance the paper's motivation for using the editing method.

---

> > ### Author Response · Authors · 2023-11-21
> > **Discussion**
> >
> > Thank you very much for the discussion and helping us to improve our work.
> >
> > **Search algorithms.** This is true that search algorithms such as beam search and MCTS can use value estimators. However, GFlowNet can be trained without access to such a value estimator. In fact, naive search algorithms need a proxy model to evaluate the unseen sequences while GFlowNet can be trained on only offline labeled dataset. For example, MCTS is a model-free reinforcement learning algorithm that does not explicitly use offline data for training a policy. Instead, it learns by iteratively simulating sequences of actions and updating its search tree based on the outcomes of these simulations. In sequence design, simulation means generating new unseen sequences and the outcome is property estimation that should be done by a proxy model. Therefore, the success of naive search algorithms depends on the availability of a reliable proxy model which may not be the case in the biological sequence domain.
> >
> > **Baselines.** We agree that there are numerous ways to generate edited sequences. However, it is not trivial how various ideas can be employed effectively for sequence editing and this may need in-depth studies in terms of algorithm design. We would like to clarify that we compare the proposed algorithm with all existing biological sequence editing methods in the literature. We analyze the performance of the proposed algorithm in terms of upper and lower bounds of edit amounts (Theorems 2 and 4), lower bound of expected reward (Theorem 1) and upper bound of property improvement (Theorem 3). This examination highlights the influence of hyperparameters on algorithm performance. These theoretical guarantees distinguish this work from other possible naive ideas.
> >
> > Also, please note that in our experimental studies we explore various scenarios. For example, another aspect that distinguishes the proposed GFNSeqEditor from other methods is that other methods are not able to combine multiple sequences to create a new sequence. In section 4.3, the proposed algorithm combines sequences to generate a new sequence similar to its parent sequences. In section 4.3, we discuss that this can also be used for sequence length reduction which has important applications in vaccine therapies.
> >
> > ## Comparison with de novo methods
> > We would like to clarify that the biological sequence design problem is different from biological sequence editing and these two problems can have different applications. In biological sequence design, the goal is to discover novel sequences with desired properties. Sequence editing, on the other hand, often refers to the modification of existing DNA, RNA, or protein sequences. Biological sequence editing has applications in creating genetically modified organisms for research purposes (https://academic.oup.com/femsyr/article/22/1/foac033/6626025) and developing potential therapeutic interventions (https://www.nature.com/articles/s41581-022-00636-2 ). Techniques like CRISPR-Cas9 enable precise editing of specific genomic sequences.
> >
> > Although GFNSeqEditor is designed for sequence editing, it can also be utilized for sequence generation. To address your concern, we employed GFNSeqEditor to generate new sequences and we compared the performance of GFNSeqEditor in sequence generation task with that of GFlowNet and Diffusion model (DM) on CRE dataset. We did this by relaxing the hyperparameters to allow a higher amount of edits and we set $\delta = 0.4$, $\lambda=0.1$ and $\sigma = 0.001$ for GFNSeqEditor. The results are presented in the Table below. GFlowNet and DM generate 1000 sequences. GFNSeqEditor also generates 1000 sequences by editing each of 1000 samples in the test dataset. As can be seen, GFNSeqEditor achieves higher property than both GFlowNet and DM. It is useful to note that the experimental study by Jain et al., (2022) have shown that GFlowNet outperforms state-of-the-art sequence design methods. For each sequence generated by GFlowNet and DM, the distance to the test set is measured as the distance between the generated sequence and its closest counterpart in the test set. On average, the distance between sequences generated by GFlowNet and the test set is 54.34%, while for DM, it is 63.59%. GFNSeqEditor achieves superior performance by editing, on average, 47.34% of a sequence in the test dataset. The distance between test set and generated sequences by GFlowNet and DM cannot be controlled. As it is studied in Figures 2 and 3 in the paper, the amounts of edits performed by GFNSeqEditor can be controlled by hyperparameters $\delta$, $\lambda$ and $\sigma$. We can add these results to the paper.
> >
> > Algorithms | Property | Diversity | Distance |
> > :--- | :---: | :---: | :---:
> > DM | 1.75 | 107.38 | 63.59%
> > GFlowNet | 28.20 | 83.88 | 54.41%
> > GFNSeqEditor | **29.25** | 87.32 | **47.34**%

---

> > > ### Comment · Reviewer_aGg2 · 2023-11-22
> > >
> > > I concur with your points regarding the comparison between different search methods. You rightly note that many other search techniques rely on "proxy" models, whereas GFlowNet distinguishes itself by learning policy directly. It's worth mentioning that MCTS can also be considered a model-based reinforcement learning method, as demonstrated in the case of AlphaGo. Thus, the question of whether model-free GFlowNets-based editing is superior to model-based search methods remains open. To address this, I recommend conducting more extensive experiments to bolster your evidence and strengthen your argument.
> > >
> > > I commend the inclusion of new results for comparison with De Novo methods, even if the comparison is limited to DM and GFlowNets.
> > >
> > > While I partially agree with your arguments, I would consider increasing my score from 5 to 6.

---

> > > > ### Author Response · Authors · 2023-11-22
> > > >
> > > > We would like to express our sincere gratitude for your constructive discussions and suggestions. Your thoughtful comments and discussions truly helped us to improve the quality of our work.
> > > >
> > > > We try our best to perform experiments to compare GFlowNets and MCTS and add the results to the paper as soon as possible. Please note that in general relying on a proxy model may not be a good practice due to possible inaccuracies in the predictions. Both directed evolution (DE) and Ledidi suffer from relying on a proxy model and in the experiments section we see that GFNSeqEditor outperforms DE and Ledidi.
> > > >
> > > > We update the paper and we add the comparison with De Novo methods to Appendix C.4 due to space limit in the main text. We refer the readers in the main text to Appendix C.4 to see these comparisons. Specifically, at the bottom of page 7 we add that “*In addition to sequence editing, the proposed GFNSeqEditor is able to generate new sequences. The performance of GFNSeqEditor in generating new sequences is studied in Appendix C.4*.”
> > > >
> > > > Thank you again for helping us to enhance our paper.

---

### Official Review · Reviewer_1B72 · 2023-11-01

**Soundness:** 2 fair
**Presentation:** 2 fair
**Contribution:** 2 fair
**Rating:** 5
**Confidence:** 2

**Summary:**

* In this paper, the authors present a novel sequence editing method that leverages GFlowNet. This method relies on a pre-trained flow function to evaluate the potential for substantial property improvement within a given sequence. Furthermore, it generates a variety of edits using a stochastic policy.
* The properties of the edited sequences are analyzed by assessing the lower and upper bounds of the reward function.
* To evaluate the effectiveness of this approach, the authors conducted real data experiments and compared their method to three baseline approaches. They assessed performance using various metrics, including property enhancement, edit percentage, and diversity in TF binding.

**Strengths:**

The experimental results demonstrate the superiority of the proposed method across various DNA and protein sequence editing tasks. It consistently outperforms other baselines by generating sequences with fewer edits, enhanced properties, and greater diversity

**Weaknesses:**

* Lack of Training Details: The paper lacks sufficient information regarding the training process of the policy. It should provide more details on the training data used, the methodology for updating parameters, and the specific hyperparameters employed in the process.
* Unclear Literature Review: The literature review in the paper needs improvement. It is not adequately clear what the main contribution of the proposed method is, and how it distinguishes itself from existing work, particularly in relation to the utilization of GFlowNet for sequence generation. The paper should provide a more explicit and comparative analysis of related work.
* Ambiguity in Key Innovation: The claim that GFNSeqEditor can produce novel sequences with improved properties lacks clarity regarding the key innovation driving these contributions. The paper should better articulate what novel techniques or insights lead to the claimed improvements, thereby enhancing the reader's understanding of the method's unique value.

**Questions:**

See the comments in Weakness section.

---

> ### Author Response · Authors · 2023-11-17
> **Response to Comments about Innovations and Training Process**
>
> Thank you very much for taking time to review our paper and letting us know your feedback. In response to your feedback, we have extended the literature review within the introduction and incorporated additional details regarding the training process and the innovative aspects of the proposed algorithm to the experiments section. Please find below our point-by-point responses to your comments.
>
> ## Training Details
> Due to the space limit, we put the training details in Appendix C.1. To address this comment, we provide more information about the training process of the flow function in the main text of the revised paper. Please see page 7 where we add that ``*To train models associated with baselines and the proposed GFNSeqEditor, we partition each dataset into a 72% training set and an 18% validation set. The remaining 10% constitutes the test set, employed to evaluate the performance of methods in sequence editing tasks. The trained flow function $F_\theta(\cdot)$ employed by GFlowNet-E and the proposed GFNSeqEditor, is an MLP comprising two hidden layers, each with a dimension of $2048$, and $|\mathbb A|$ outputs corresponding to actions. Throughout our experiments, we employ the trajectory balance objective for training the flow function. Detailed information about training the flow function can be found in Appendix C.1.*''  This revision is highlighted in red in the revised version. Also, we expand the implementation details in Appendix C.1.
>
> ## Key Innovation:
> We would like to emphasize that the primary goal of GFNSeqEditor is not to produce novel sequences and the focus is different from biological sequence design. While GFNSeqEditor can indeed be utilized in conjunction with biological-sequence generation models to create novel sequences with improved properties, as empirically examined in section 4.2, the primary intent of GFNSeqEditor is not centered around generating entirely novel sequences. Instead, the core purpose of the proposed GFNSeqEditor lies in generating a diverse array of edits for a given seed sequence $x$, ensuring that the edited sequences exhibit similarity to $x$ while simultaneously displaying improved properties. To highlight the key innovations that contributed to the performance gain of GFNSeqEditor compared to other baselines, in this revised version we expand the discussion about the results in Table 1 and Figure 3. We explain that the proposed sub-optimal identification method outlined in equation 7 and the proposed editing policy in equation 9 are the key innovations to achieve improved performance compared to other baselines. This revision is highlighted in red on page 7 in section 4.1.

---

> ### Author Response · Authors · 2023-11-17
> **Response to Comment about Literature Review**
>
> ## Literature Review
> To address this comment, we expand the literature review in the introduction and we discuss our contribution relative to the biological sequence design methods including GFlowNet. This revision can be found on pages 1 and 2 highlighted in red. Also we extend the detailed related work discussion in Appendix D. Please find below the discussion about the contributions that distinguish this paper from prior works.
>
> ### Biological Sequence Design
> Existing machine learning methods within the domain of biological sequences have predominantly concentrated on generating novel sequences with desired properties from scratch. Therefore, there is an inherent risk of deviating significantly from naturally occurring sequences compromising safety and predictability. In contrast, this paper addresses the generation of new edited sequences that enhance the properties of existing sequences, ensuring the edited ones maintain similarity to their existing counterparts. Here, we take an input seed sequence and modify a few elements to improve its property. To underscore the limitations of existing biological sequence design methods, we conducted experiments on AMP and CRE datasets using GFlowNet to generate sequences from scratch. Our observations reveal that on average GFlowNet-generated sequences for the CRE dataset differ from existing sequences at 76.29% of locations, and for the AMP dataset, the difference is found in 95.24% of locations. In contrast, the proposed GFNSeqEditor demonstrates the ability to generate new edited sequences differing from existing ones in less than 35% of locations across all datasets. Moreover, employing existing biological sequence methods the difference between generated sequences and existing ones cannot be controlled while using the proposed GFNSeqEditor the amount of edits can be controlled by hyperparameters.
>
> ### Sequence Combination
> Also another aspect that distinguishes the proposed GFNSeqEditor from prior works is that existing methods are not able to combine multiple sequences to create a new sequence. In section 4.3, the proposed algorithm combines sequences to generate a new sequence similar to its parent sequences. In section 4.3, we discuss that this can be used for sequence length reduction which has important applications such as vaccine therapies.
>
> ### GFlowNet
> Existing GFlowNet methods can be used for biological sequence design from scratch and as a result, they suffer from the same problems as those of other biological sequence design methods discussed above when it comes to performing sequence editing. However, since GFlowNet can generate sequences token by token sequentially, there is a naive approach to employ GFlowNet for sequence editing. In this approach, the first tokens of a sequence $x$ can be given to GFlowNet and then the GFlowNet generates the rest of the tokens. We add this as a baseline to Table 1 in the revised version and the baseline is called GFlowNet-E. For example, for the CRE dataset, the GFlowNet-E gets the first 130 tokens and generates the rest of 70 tokens. Compared to GFNSeqEditor, this approach lacks the flexibility to choose edit locations and GFNSeqEditor outperformed GFlowNet-E. This indicates the effectiveness of sub-optimal position identification of GFNSeqEditor.

---

> > ### Comment · Reviewer_1B72 · 2023-11-21
> > **Reply to authors**
> >
> > Thanks to the authors for the clarification and supplemental details. I've updated my score accordingly.

---

> > > ### Author Response · Authors · 2023-11-22
> > >
> > > Thank you very much for your feedback on our responses to your comments. As we approach the end of the discussion period, we are taking this opportunity to inquire if you have any specific concerns or queries regarding our paper. Thank you for your help.

---

### Comment · Area_Chair_Vdi5 · 2023-11-18
**Please take a look at authors' comments**

The authors have compiled a few responses. Please take the time to look at these responses as early as possible.

---

### Meta-Review · Area_Chair_Vdi5 · 2024-01-02

**Metareview:**

The paper proposes an approach to edit a set of candidate biological sequences for a given task using GFlowNets. The idea is to revisit some of the transition probabilities of a set of candidates, and detect suboptimal locations to flip (randomly, with an accept/reject mechanism) / edit letters. These rules depend therefore on spotting whether there is an opportunity to flip ($\delta$ to select suboptimal gaps, $\sigma$ for randomness) and, when the decision is taken to redraw a sample, whether to over-under penalize that new change (parameterized by a low $\lambda$). Algorithm (see Alg. 1) can be seen as exploiting the knowledge accumulated in a pretrained GFlowNet model to mutate sequences efficiently.

The paper was appreciated by all reviewers, whose score distribution moved from 3,5,5 to 5,5,6. While this is a great sign that the authors have provided a nice and constructive rebuttal, the lack of additional support from any of the reviewers, coupled with the fact that a big chunk of the criticism is (still, after rebuttal) directed towards the experimental part of the paper (all reviewers agree that the core idea is nice, but simple) make me think the paper is on the reject side of the fence at ICLR. The constructive criticism provided by reviewers should help authors improve their submission.

**Justification For Why Not Higher Score:**

The method seems quite basic, and experiments have been criticized for not adding more fundamental baselines (including GFlowNet).

**Justification For Why Not Lower Score:**

NA

---

### Decision · Program_Chairs · 2024-01-16

Reject